# Tumor-Derived Extracellular Vesicles Inhibit Natural Killer Cell Function in Pancreatic Cancer

**DOI:** 10.3390/cancers11060874

**Published:** 2019-06-22

**Authors:** Jiangang Zhao, Hans A. Schlößer, Zhefang Wang, Jie Qin, Jiahui Li, Felix Popp, Marie Christine Popp, Hakan Alakus, Seung-Hun Chon, Hinrich P. Hansen, Wolfram F. Neiss, Karl-Walter Jauch, Christiane J. Bruns, Yue Zhao

**Affiliations:** 1Department of General, Visceral und Tumor Surgery, University Hospital Cologne, Kerpener Straße 62, 50937 Cologne, Germany; jiangang.zhao@uk-koeln.de (J.Z.); hans.schloesser@uk-koeln.de (H.A.S.); zhefang.wang@uk-koeln.de (Z.W.); jie.qin@uk-koeln.de (J.Q.); jiahui.li@uk-koeln.de (J.L.); felix.popp@uk-koeln.de (F.P.); marie.popp@uk-koeln.de (M.C.P.); hakan.alakus@uk-koeln.de (H.A.); seung-hun.chon@uk-koeln.de (S.-H.C.); christiane.bruns@uk-koeln.de (C.J.B.); 2Department of General, Visceral und Vascular Surgery, Ludwig-Maximilian-University (LMU), 81377 Munich, Germany; Karl-Walter.Jauch@med.uni-muenchen.de; 3Department I of Internal Medicine, University Hospital of Cologne, Center for Integrated Oncology Cologne-Bonn, CECAD Center of Excellence on “Cellular Stress Responses in Aging-Associated Diseases”, Center for Molecular Medicine Cologne, University of Cologne, 50931 Cologne, Germany; hinrich-peter.hansen@uk-koeln.de; 4Department of Anatomy I, Medical Faculty, University of Cologne, 50937 Cologne, Germany; wolfram.neiss@uk-koeln.de; 5Department of General, Visceral und Vascular Surgery, Otto von Guericke University, 39120 Magdeburg, Germany

**Keywords:** extracellular vesicles, NK cells, pancreatic cancer, pre-metastatic niche

## Abstract

Pancreatic ductal adenocarcinoma (PDAC) is one of the most lethal malignancies. Tumor-derived extracellular vesicles (EVs) induce pre-metastatic niche formation to promote metastasis. We isolated EVs from a highly-metastatic pancreatic cancer cell line and patient-derived primary cancer cells by ultracentrifugation. The protein content of EVs was analyzed by mass spectrometry. The effects of PDAC-derived EVs on natural kill (NK) cells were investigated by flow cytometry. The serum EVs’ TGF-β1 levels were quantified by ELISA. We found that integrins were enriched in PDAC-derived EVs. The expression of NKG2D, CD107a, TNF-α, and INF-γ in NK cells was significantly downregulated after co-culture with EVs. NK cells also exhibited decreased levels of CD71 and CD98, as well as impaired glucose uptake ability. In addition, NK cell cytotoxicity against pancreatic cancer stem cells was attenuated. Moreover, PDAC-derived EVs induced the phosphorylation of Smad2/3 in NK cells. Serum EVs’ TGF-β1 was significantly increased in PDAC patients. Our findings emphasize the immunosuppressive role of PDAC-derived EVs and provide new insights into our understanding of NK cell dysfunction regarding pre-metastatic niche formation in PDAC.

## 1. Introduction

Pancreatic ductal adenocarcinoma (PDAC) is one of the most lethal malignancies worldwide [1]. The majority of patients with pancreatic cancer are diagnosed at an advanced stage without the opportunity for curative surgery [2]. Even after R0 resection followed by adjuvant chemotherapy and/or radiotherapy, most patients will eventually suffer from recurrence [3]. The Surveillance, Epidemiology and End Results (SEER) Program (https://seer.cancer.gov/statfacts/) revealed that the five-year overall survival for PDAC patients with localized disease is 34.3%. For those who present with distant metastases, this drops to merely 2.7%. These daunting statistics emphasize the importance of improving our understanding of the metastatic process to reduce the incidence of metastasis and develop effective therapeutic strategies for PDAC patients.

As an important hallmark of cancer, metastasis is a complex process that propagates tumor cells from the site of origin to distant tissues, also known as the metastatic cascade [4,5]. The term “pre-metastatic niche” describes the microenvironment in a secondary organ that has been affected by the primary tumor to support tumor growth in advance of tumor cell entry [6]. Many studies have identified the existence of pre-metastatic niches in different organs, such as liver and lung [7,8]. Liu Y. et al. summarized six characteristics for the pre-metastatic niche, including immunosuppression, inflammation, lymphangiogenesis, angiogenesis/vascular permeability, organotropism, and reprogramming [9]. The liver pre-metastatic niche has been partly analyzed in orthotopic PDAC mouse models [7]. However, the underlying mechanisms of liver pre-metastatic niche formation remain unclear in human PDAC. Extracellular vesicles (EVs), like exosomes and microvesicles, can mediate intercellular communication, and thus affect physiological and pathological conditions through the transfer of various cargo molecules, including proteins, nucleic acids, and lipids [10,11]. As lipid bilayer membrane vesicles, EVs are one ideal carrier for drug delivery in cancer treatment [12,13]. Kamerkar S et al. have successfully modified exosomes to deliver short interfering RNA specific to KRAS mutation, which suppressed tumor growth in PDAC-bearing mice and significantly increased their overall survival [14]. Jang SC et al. synthesized nanovesicles by the breakdown of monocytes or macrophages, which had similar characteristics as the exosomes. These nanovesicles could deliver chemotherapeutics to inhibit tumor growth [15]. Saari H and team have loaded Paclitaxel into cancer cell-derived EVs. Paclitaxel-loaded EVs could bring the drug into prostate cancer cells and increase its cytotoxicity [16]. In addition, a study led by Garofalo et al. reported that human lung cancer cell-derived extracellular vesicles (EVs) formulations could specifically target the neoplasia. Administration of EVs with oncolytic virus alone or in combination with chemotherapeutics may serve as a novel strategy to treat cancer [17].

In addition, tumor-derived EVs can be released into the circulation and transferred to distant sites. Recent research has shed light on the role of tumor-derived EVs in pre-metastatic niche formation. In murine models of lung cancer and melanoma, tumor-derived exosomes delivered signals to lung epithelial cells and activated Toll-like receptor 3, which elicited chemokine production and promoted neutrophil infiltration [8]. PDAC-derived exosomes recruited macrophages and neutrophils to the liver and stimulated hepatic stellate cells to synthesize fibronectin to promote liver metastasis [7]. In addition, exosomal integrins are thought to determine organotropic metastasis [18]. In conclusion, tumor-derived exosomes are involved in pre-metastatic niche formation, such as angiogenesis, immunosuppression, and organotropism.

Natural killer (NK) cells are a group of innate lymphocytes with inherent capacities to recognize and eliminate virus-infected cells and malignant cells. The role of NK cells in PDAC has received less attention, but is increasingly being recognized. Gürlevik E et al. reported that after primary tumor resection, gemcitabine treatment triggered NK cell cytotoxicity against pancreatic cancer cells and decreased the incidence of local recurrence in orthotopic PDAC mouse models [19]. Ames E et al. found that NK cells preferentially killed pancreatic cancer stem cells (CSCs) in vitro. Intratumoral injection of NK cells in human pancreatic cancer-bearing NSG mice reduced the percentage of pancreatic CSCs and the tumor burden [20]. However, the ability to escape from immune surveillance has been established as a hallmark of tumor cells [5]. Increasingly, studies have revealed NK cells’ exhaustion in tumor. Exhausted NK cells produce decreased cytokines, downregulate activating receptors, as well as exhibit impaired cytolytic activity [21,22,23]. In addition, dysregulated cellular metabolism has been observed in dysfunctional NK cells. Cong J et al. found that in lung cancer mouse models, the expression of gluconeogenesis enzyme fructose bisphosphatase 1 (FBP1) was upregulated in tumor-infiltrating NK cells, which mediates the dysfunction of NK cells by impairing glycolysis [24]. The mechanisms for NK cell dysfunction include direct inhibition via cell–cell contact and indirect inhibition via the production of inhibitory factors such as transforming growth factor beta 1 (TGF-β1), interleukin 10 (IL-10), prostaglandin E2 (PGE2), and indoleamine 2,3-dioxygenase (IDO) [25,26]. However, to date, the dynamic status of NK cells in the liver pre-metastatic niche of PDAC remains unclear. Tumor-derived EVs are emerging as a regulator in NK cell modulation in pancreatic cancer.

In this study, we have investigated the effect of tumor-derived EVs on NK cells in pancreatic cancer. Here, we provide evidence that pancreatic cancer-derived EVs carry abundant immunosuppressive factors and inhibit NK cell function, which contributes to pre-metastatic niche formation.

## 2. Results

### 2.1. Characterization of Pancreatic Cancer-Derived EVs

EVs were isolated from cell culture supernatants of a highly-metastatic pancreatic cancer cell line L3.6pl and a PDAC patient-derived primary cancer cell line TBO368 by differential centrifugation and ultracentrifugation (Appendix A). In order to examine the morphology and measure the size of pancreatic cancer-derived EVs, we used transmission electron microscopy (TEM) and nanoparticle analysis (NTA). The TEM image showed that pancreatic cancer-derived EVs displayed features of membrane vesicles (Figure 1a andAppendix A). The result of NTA demonstrated that most of the pancreatic cancer-derived EVs had a diameter of around 110 nm (Figure 1b). EVs were further characterized by their expression of CD9, CD63, CD81, ALIX, flotillin-1, TSG101, and Rab5, which are commonly-used markers for EVs (Figure 1c,d and Appendix A. Intriguingly, we also detected mutant KRAS (G12D) in L3.6pl-derived EVs, which was consistent with that in genomic DNA (Appendix A).

### 2.2. Comprehensive Proteomic Analysis of Pancreatic Cancer-Derived EVs

The proteomic profile of pancreatic cancer-derived EVs was analyzed by mass spectrometry. More than 2600 proteins were detected in both samples. A significant overlap was observed in L3.6pl-derived EVs and TBO368-derived EVs. (Appendix A). Eighty-eight of the top 100 most frequently-identified proteins in EVs, according to the Exocarta database (www.exocarta.org), were identified in pancreatic cancer-derived EVs. The enrichment of exosomal markers verified the purity of EVs (Appendix A). 

To investigate the cellular component, molecular function, and biological process of proteins in pancreatic cancer-derived EVs, GO analysis was performed using the Gene Ontology Resource (http://geneontology.org/). Proteins were categorized according to their ontology as determined from their GO annotation terms. Based on the cellular component, around 40% of all the identified proteins were annotated to extracellular EVs (Figure 2a). The molecular function revealed the enrichment of proteins related to translation regulator activity (GO:0045182), transcription regulator activity (GO:0140110), molecular transducer activity (GO:0060089), binding (GO:0005488), structural molecule activity (GO:0005198), molecular function regulator (GO:0098772), catalytic activity (GO:0003824), and transporter activity (GO:0005215) (Figure 2b). The biologic process revealed that the proteins in pancreatic cancer-derived EVs were involved in cellular component organization or biogenesis (GO:0071840), cellular processes (GO:0009987), biological phases (GO:0044848), localization (GO:0051179), reproduction (GO:0000003), biological regulation (GO:0065007), response to stimulus (GO:0050896), developmental processes (GO:0032502), multicellular organismal processes (GO:0032501), biological adhesion (GO:0022610), metabolic processes (GO:0008152), cell proliferation (GO:0008283), and immune system processes (GO:0002376) (Figure 2c).

### 2.3. Pancreatic Cancer-Derived EVs Carry Adhesion Molecules

To evaluate the role of pancreatic cancer-derived EVs in the pre-metastatic niche, GO analysis revealed that abundant cellular adhesion proteins exist in pancreatic cancer-derived EVs, particularly integrins, such as ITGA1, ITGA2, ITGA3, ITGA6, ITGAV, ITGB1, ITGB4, ITGB5, ITGB6, and ITGB8 (Figure 3a,b). We detected the expression of integrin alpha V (ITGAV) in L3.6pl-derived EVs by Western blotting (Figure 3c). To track the *in vivo* distribution of pancreatic cancer-derived EVs, we further injected PKH67-labelled L3.6pl-derived EVs intravenously into NOD-scid IL2rγ^null^ (NSG) mice. Twenty four hours after injection, PKH67-labelled EVs were detected by immunofluorescence in cryosections of mouse liver tissue, which indicated that pancreatic cancer-derived EVs reached the liver (Figure 3d and Appendix A).

### 2.4. Pancreatic Cancer-Derived EVs Carry Immune Regulatory Factors

To investigate the role of tumor-derived EVs in immune regulation, we first analyzed the expression pattern of immune regulatory factors in paired PDAC tumor tissues and adjacent non-tumor tissues based on the GSE28735 dataset (*n* = 45). Compared to non-tumor tissues (N), a variety of factors like TGF-β1, TGF-β2, HMGB1, PVR, nectin-2, galectin-9, PD-L1, PD-L2, and MICA/MICB were significantly higher in the tumor tissue (T) (Figure 4a). Interestingly, enrichment of some molecules, including TGF-β1, nectin-2, and PVR, was identified in pancreatic cancer-derived EVs by Western blotting (Figure 4b). TGFbRI and TGFbRII (TGF-β1 receptors), DNAM-1, TIGIT, and CD96 (nectin-2 and PVR receptors) are present on NK cells. These results support the hypothesis that pancreatic cancer-derived EVs potentially modulate NK cell function.

### 2.5. Pancreatic Cancer-Derived EVs Inhibit NK Cell Function

Subsequently, we determined whether NK cells could uptake pancreatic cancer-derived EVs. To address this issue, L3.6pl-derived EVs were stained with PKH67 (green). PKH67-labelled EVs were incubated with NK cells. After 24 h, we observed that PKH67-labelled EVs were present on the plasma membrane and in the cytoplasm of NK cells (Figure 4c). This result indicated that pancreatic cancer-derived EVs could be incorporated by NK cells, suggesting their potential role in the regulation of NK cell function. Thus, we examined the effects of pancreatic cancer-derived EVs on NK cells. 

NKG2D is one of the most important activating receptors on NK cells, and the expression level of NKG2D correlates positively with their anti-tumor ability [27]. We co-cultured NK cells with L3.6pl-derived EVs or PBS for 24 h. After co-culture, the expression of NKG2D in NK cells was significantly downregulated (Figure 4d). CD107a is a functional marker for NK cells [28]. TNF-α and IFN-γ are two main cytokines produced by activated NK cells [29]. To measure the amount of CD107a, TNF-α, and IFN-γ, NK cells pre-treated with L3.6pl-derived EVs or PBS were co-cultured with L3.6pl cells at an effector:target cell ratio of 1:1 for 5 h. L3.6pl-derived EVs resulted in a significant decrease of CD107a, TNF-α, and IFN-γ in NK cells (Figure 4e). Nutrient uptake and glucose metabolism are essential for NK cell functionality [30]. CD71 (transferrin receptor), CD98 (large neutral amino acid transporter), and 2-NBDG incorporation ability are three commonly-used metabolic parameters in NK cells [31]. We found that L3.6pl-derived EVs significantly reduced the expression of CD71 and CD98 in NK cells. In addition, L3.6pl-derived EVs impaired the glucose uptake ability of NK cells (Figure 4f).

### 2.6. Pancreatic Cancer-Derived EVs Impair NK Cell Cytotoxicity

It has been reported that NK cells have the capacity to kill CSCs [20]. We next examined whether pancreatic cancer-derived EVs impair NK cell cytotoxicity against pancreatic CSCs. After enrichment of CSCs using the sphere formation assay, we found higher mRNA expression levels of NKG2D ligands, MICB, and ULBP2 in the CSCs population (Figure 5a). Flow cytometric analysis confirmed a higher MICA/MICB expression in spheres than that in adherent cells (Figure 5b). This indicated that NK cells might prefer to recognize and eliminate pancreatic CSCs. NK cells were then pre-treated with L3.6pl-derived EVs in the presence of IL-2 (100 U/mL) for 24 h. Then, we co-cultured L3.6pl cells with untreated or L3.6pl-derived EV pre-treated NK cells. After 24-h killing, floating cells were washed away, and adherent cells were trypsinized for the sphere formation assay. We found that NK cells pre-treated with L3.6pl-derived EVs showed decreased cytotoxicity against pancreatic CSCs (Figure 5c,d).

### 2.7. Pancreatic Cancer-Derived EVs Phosphorylate Smad2/3 in NK Cells

As a major immunosuppressive cytokine, TGF-β1 inhibits the activation and function of NK cells through the TGFβ-Smad2/3 signaling pathway [32]. In our experiments, we observed that TGF-β1 attenuated the expression of the NKG2D, CD107a, IFN-γ, CD71, CD98, and 2-NBDG incorporation ability of NK cells (Appendix A). As shown above, pancreatic cancer-derived EVs contained TGF-β1. Therefore, we investigated whether pancreatic cancer-derived EVs could activate the TGFβ-Smad2/3 signaling pathway in NK cells. After incubation with TGF-β1 or L3.6pl-derived EVs, the phosphorylation level of Smad2/3 in NK cells was significantly elevated. However, in the presence of SB-431542 (an inhibitor of TGFβRI), the phosphorylation of Smad2/3 was reversed and returned to the original baseline level (Figure 6a). These findings suggest that pancreatic cancer-derived EVs deliver TGF-β1 to NK cells, induce Smad2/3 phosphorylation, and ultimately result in NK cell dysfunction.

### 2.8. Increased TGF-β1 Levels in Serum EVs of Patients with PDAC

The clinicopathological characteristics of patients with PDAC are listed in Appendix A. The mean age of the patients was 66.1 y, and they were predominantly male. All the patients donated blood at the time point of diagnosis prior to any treatment (*n* = 30). At diagnosis, 53.3% of patients presented with a tumor stage T1 and T2, and 46.7% presented with T3 or T4. Seventy seven-point-seven percent of patients had a positive lymph node status. Three out of 30 patients had distant metastases (M1). Fifty six-point-seven percent of the patients were Union for International Cancer Control (UICC) I or II, and 43.3% were UICC III or IV. Nineteen healthy individuals were included in this study as controls (*n* = 19). TGF-β1 was overexpressed in PDAC (Figure 6b). The amount of TGF-β1 in serum EVs was determined by ELISA. The concentration of TGF-β1 per gram of EVs was calculated. The level of EVs’ TGF-β1 in patients with pancreatic cancer ranged from 0.20–0.88 ng/g. In the healthy donors, it ranged from 0.05–0.30 ng/g. Compared to healthy donors, TGF-β1 in serum EVs was significantly elevated in patients with PDAC (*p* < 0.0001) (Figure 6c).

## 3. Discussion

Pancreatic ductal adenocarcinoma is one of the most lethal malignancies [1]. Metastasis accounts for a majority of cancer-related deaths in PDAC. Recently, the pre-metastatic niche has been proposed to elucidate the mechanisms of the organ-specific metastatic process in many cancer entities, such as melanoma, lung cancer, and pancreatic cancer [7,8,33]. Over the past few decades, EVs have attracted wide attention in early detection, diagnosis, and treatment of cancer. Moreover, as a mediator of intercellular communication, EVs released from tumor cells are found to interact with cells in distant organ sites and finally induce a pre-metastatic niche for future metastasis [34]. Despite tremendous advances, the underlying cellular and molecular events involved in pre-metastatic niche formation of PDAC have yet to be determined.

In the present study, tumor-derived EVs were isolated from two human pancreatic cancer cell lines, L3.6pl and TBO368, by differential centrifugation and ultracentrifugation to exclude dead cells, large debris, and microvesicles. Then, the morphology and size distribution of EVs were examined by TEM and NTA. Exosomal markers, including CD9, CD63, CD81, TSG101, Alix, glotillin-1, and Rab5, were identified by Western blotting. Comprehensive proteomic analysis is expected to elucidate the potential impact of tumor-derived EVs on pre-metastatic niche formation of PDAC. The proteomic profile of mouse PDAC cell line-derived EVs has been analyzed by a previous study. Yu Z et al. compared exosomes derived from Panc02 and Panc02-H7 cells using proteomic analyses. The differentially-expressed proteins in Panc02-H7-derived exosomes were thought to enhance tumor growth, invasion, and metastasis [35]. To the best of our knowledge, this is the first study to systematically analyze the protein content in human PDAC-derived exosomes. By mass spectrometry, more than 2600 proteins were detected in both samples. We found that about 90% of the protein identified in L3.6pl-derived EVs overlapped with those identified in TBO368-derived EVs. GO analysis of identified proteins was performed for cellular components, molecular functions, and biologic processes.

GO-based category clustering of the molecular functions of protein contents in pancreatic cancer-derived EVs suggested that there was a significant enrichment in localization and biological adhesion, which may facilitate the ability of EVs’ to adhere to the surfaces of recipient cells, fuse with their membranes, and transfer exosomal components into the target cells to modulate their biological functions. The mechanisms of organ-specific homing and colonization of cancer cells are enormously complex. Cell adhesion to the extracellular matrix (ECM) determines the colonization of metastatic sites and facilitates the survival of circulating tumor cells in the new environment. Integrins can bind to fibronectin, vitronectin, laminin, and collagen in the ECM, thereby enhancing tumor cell motility and invasion ability [36]. Y Liu et al. reported that after either intravenous injection or intra-tumor injection, lung cancer-derived exosomes were detected in the lung and induced a lung pre-metastatic niche in mouse models [8]. GO analysis revealed that pancreatic cancer-derived exosomes exhibited abundant cellular adhesion molecules, especially integrins. Ayuko Hoshino et al. found that tumor-derived exosomal integrins determined organotropic metastasis. They reported that tumor-derived exosomes carrying integrins α6β4 and α6β1 were responsible for lung metastasis, while exosomes carrying integrin αvβ5 were associated with liver metastasis [18]. Our findings showed that integrin αv and integrin β5 were abundant in pancreatic cancer-derived EVs. After intravenous injection, PKH67-labeled pancreatic cancer-derived EVs reached the liver of the NSG mouse. Therefore, we proposed that tumor-derived EVs’ tended to enter the liver, delivered cargos to the recipient cells, and induced a pre-metastatic niche for future metastasis in PDAC.

A key feature of the pre-metastatic niche is immunosuppression [9]. In our study, GO analysis revealed that the identified proteins in pancreatic cancer-derived EVs were involved in biological regulation and immune system processes. By Western blotting, we found that pancreatic cancer-derived EVs displayed a variety of immune regulatory molecules, such as TGF-β1, nectin-2, and PVR. Therefore, we speculated that pancreatic cancer-derived EVs might be involved in the modulation of immune cell functions. Next, we tried to investigate the effects of pancreatic cancer-derived EVs on NK cells. Firstly, we found that pancreatic cancer-derived EVs could be incorporated by NK cells. Nevertheless, it was still uncertain whether pancreatic cancer-derived EVs could mediate immune suppression upon co-incubation with NK cells.

The stress proteins MICA and MICB are widely expressed by cancer cells due to genomic damage [37]. NKG2D is a key activating receptor for NK cell cytotoxicity [27]. The binding of MICA/MICB to NKG2D receptors triggers NK cell-mediated cytotoxicity and enables them to eliminate cancer cells [38]. Our results indicated that the expression of NKG2D on NK cells was significantly downregulated by pancreatic cancer-derived EVs. After recognition and activation, NK cells synthesize and release effective cytokines into the tumor cells. For example, IFN-γ and TNF-α are two indispensable cytokines for NK cell cytotoxicity. Neutralization of IFN-γ and TNF-α significantly impaired NK cell activity [29]. We demonstrated that treatment with EVs led to less production of IFN-γ and TNF-α in NK cells. In addition, as a functional marker for NK cell activity, the expression of CD107a was significantly downregulated.

Recently, the importance of cellular metabolism of immune cells has gained increasing attention. Enough nutrients and energy are essential for NK effector functions [39]. Cong J et al. demonstrated that the tumor could reduce NK cell glycolytic capacity, which results in reduced cytotoxicity and NK cell dysfunction [24]. We found that NK cells exhibited less CD71 and CD98, as well as reduced glucose uptake ability after treatment with EVs. Dysregulated metabolism caused by pancreatic cancer-derived EVs affected multiple biological processes in NK cells such as interfered protein synthesis and impaired energy production. Our result was consistent with previous studies investigating the effects of tumor-derived EVs on immune cells [40,41,42]. Our findings suggest that pancreatic cancer-derived EVs induce a dysfunctional phenotype of NK cells, which ultimately contribute to an immunosuppressive microenvironment in the pre-metastatic niche.

Pancreatic cancer cells with high aldehyde dehydrogenase 1 (ALDH1) expression are considered as cancer stem cells (CSCs) [43]. Here, we found that after enrichment using the sphere formation assay, pancreatic CSCs exhibited high expression of ligands for NKG2D. However, pancreatic cancer-derived EVs impaired NK cell cytotoxicity against CSCs. CSCs are thought to be responsible for metastasis [44]. The inhibition of NK cell cytotoxicity allowed pancreatic CSCs to escape from NK cell immune surveillance and colonize in the target organ.

As a key inhibitory cytokine, TGF-β1 plays a dominant role in modulating NK cell function. For example, TGF-β1 attenuated NK cell responses by downregulating NKG2D expression in patients with advanced cancer [45]. Among various signals delivered by pancreatic cancer-derived EVs to NK cells, TGF-β1 was thought to be a candidate responsible for NK cell dysfunction. Our results showed that TGF-β1 impaired NK cell function, including downregulated expression of NKG2D, CD107a, CD71, and CD98, decreased production of cytokines, such as TNF-α and IFN-γ, as well as reduced glucose uptake ability. In general, activation of the TGF-β/Smad2/3 signaling pathway is implicated in NK cell dysfunction [32]. We found that either pancreatic cancer-derived EVs or TGF-β1 could induce the phosphorylation of Smad2/3 in NK cells. However, the phosphorylation level of Smad2/3 returned to baseline in the presence of SB-431542. Therefore, we proposed that pancreatic cancer-derived EVs inhibit NK cell function via the TGFβ1-Smad2/3 pathway. Pancreatic cancer-derived EVs delivered TGF-β1 to the surface of NK cells upon binding to the TGFβ receptors (TGFβRI/II). Activation of TGFβRI/II by TGF-β1 induced the phosphorylation of serine/threonine residues and triggered phosphorylation of Smad2/3. Phosphorylated-Smad2/3 then translocated to the nucleus and regulated gene transcription, thereby modulating NK cell function [46]. The TGFβRI inhibitor, SB-431542, exhibited cytotoxicity against pancreatic cancer cells in vitro [47]. Further investigations are needed to explore its anti-tumor effect in vivo, especially its influence on the phenotypic and functional diversity of NK cells.

Without specific symptoms, it is a major challenge to detect PDAC at early stages. EVs are promising to be developed as a liquid biopsy tool for early detection and diagnosis of PDAC [48]. Recently, it has been reported that the levels of exosomal PD-L1 in the plasma, rather than soluble PD-L1, were associated with disease progression in patients with head and neck squamous cell carcinomas (HNSCCs) [49]. In our study, we measured the levels of serum EVs’ TGF-β1 in PDAC patients. Interestingly, compared to healthy donors, serum EVs’ TGF-β1 was significantly elevated in the PDAC group. Therefore, serum EVs’ TGF-β1 holds promise to be used as a diagnostic tool for detection of PDAC.

Apart from TGF-β1, pancreatic cancer-derived EVs also contained multiple other immune regulatory factors, such as PVR and nectin-2, which could be delivered as inhibitory signals to NK cells. Both PVR and nectin-2 can bind to inhibitory receptors on NK cell, including CD96, PVRIG, and TIGIT [50]. Therefore, in addition to TGF-β1, PVR and nectin-2 in pancreatic cancer-derived EVs could also impair NK cell function. It has been demonstrated that the PVR/nectin-2-TIGIT axis is involved in attenuated NK cell cytotoxicity [51]. As a checkpoint, receptor blockade of TIGIT prevented NK cell dysfunction and elicited NK cell anti-tumor responses in tumor-bearing mouse models [52]. Immune checkpoint inhibitors targeting CTLA-4, PD-1, and PD-L1 have shown clinical benefit for patients with non-small cell lung cancer (NSCLC), advanced melanoma and several other cancers [53,54,55]. However, these inhibitors are less effective in patients with PDAC [56]. Our results indicate that TIGIT offers a potential immunotherapeutic target for PDAC.

Nevertheless, a limitation of the present study is that in vivo effects of human pancreatic cancer-derived EVs on NK cells have not been investigated. To address this issue, humanized patient-derived xenograft mouse models, which can better recapitulate tumor heterogeneity and simulate the complexity of the immune system, serve as a better platform for further investigation [57].

In conclusion, we proposed a novel mechanism of immune escape in PDAC. Pancreatic cancer can establish a pre-metastatic niche in the liver via tumor-derived EVs. Pancreatic cancer-derived EVs carrying immunosuppressive cargos mediate NK cell dysfunction. Metastatic pancreatic cancer cells evade the immune surveillance of NK cells and ultimately generate metastases in the liver. Additionally, serum EVs’ TGF-β1 may represent a promising non-invasive diagnostic tool for PDAC.

## 4. Materials and Methods

### 4.1. Cell Culture

The L3.6pl human pancreatic cancer cells were isolated and established after several cycles of *in vivo* selection in nude mice [58]. TBO368 was obtained with written consent from a pancreatic cancer patient and subsequently established as low-passage primary cell cultures. The study has been approved by the Ethics Committee of the University of Cologne (BIOMASOTA, (Biologische Material Sammlung zur Optimierung Therapeutischer Ansätze), ID: 13-091, approval in May 2016). L3.6pl was cultured in DMEM medium (Gibco, ThermoFisher Scientific, Waltham, MA, USA) supplemented with 10% FBS (*vol/vol*), 1% vitamin mixture, 1% sodium pyruvate, 1% nonessential amino acids, 1% L-glutamine and 1% penicillin/streptomycin (FBS from Capricorn Scientific, Ebsdorfergrund, Germany)(others from Invitrogen, ThermoFisher Scientific, Waltham, MA, USA). TBO368 was cultured in advanced DMEM medium (Gibco) supplemented with 10% FBS, 1% L-glutamine and 1% penicillin/streptomycin. Human NK cell line NK92 cells were obtained from the ATCC and cultured in MEM α, no nucleosides medium (Gibco) containing 12.5% FBS, 12.5% horse serum (Gibco), 100 IU/mL penicillin, 100 μg/mL streptomycin, 0.02 mM folic acid, 0.1 mM 2-mercaptoethanol, 0.2 mM Myo-inositol, 2 mM l-glutamine, and 100 U/mL recombinant human IL-2 (Peprotech, Hamburg, Germany).

### 4.2. EVs’ Isolation

For EVs’ preparation from cell culture supernatants, pancreatic cancer cells with a confluency of 70–80%, around 1.5 × 10^7^ cells, were washed with Dulbecco’s Phosphate-Buffered Saline (DPBS) 3 times and cultured in 25 mL of medium supplemented with 10% EVs-free FBS for an additional 24 h in a T-175 flask. EVs were isolated using a differential centrifugation and ultracentrifugation method. In brief, supernatants were centrifuged at 300× *g* for 10 min and 2000× *g* for 10 min at 4 °C to remove dead cells and cell debris. The supernatants were transferred into new tubes and centrifuged at 10,000× *g* for 30 min at 4 °C to remove large vesicles. One hundred and twenty milliliters of supernatant were then transferred into ultracentrifugation tubes and ultracentrifuged at 100,000× *g* for 70 min at 4 °C (Beckman Coulter, OptimaTM L-90K, Indianapolis, IN, USA). After the first round of ultracentrifugation, the supernatant was discarded. The pellet was resuspended in PBS and ultracentrifuged again at 100,000× *g* for 70 min at 4 °C. EVs were resuspended in 100 μL of PBS and stored in a −80 °C freezer for less than one year.

Serum EVs were isolated by a precipitation method using ExoQuick (System Biosciences, Palo Alto, CA, USA) according to the manufacturer’s instructions. In brief, serum samples were centrifuged at 3000× *g* for 15 min at room temperature (RT) to remove remaining blood cells and cell debris. After centrifugation, 250 μL of serum were put into a new tube and mixed with 63 μL of ExoQuick Exosome Precipitation Solution (System Biosciences, Palo Alto, CA, USA). The mixture was incubated at 4 °C for 30 min and then centrifuged at 1500× *g* at 4 °C for 30 min. The supernatant was aspirated. The tube was centrifuged at 1500× *g* for an additional 5 min to remove the residual ExoQuick solution. The pellet was resuspended completely in 100 μL of PBS. Serum EVs were stored in a −80 °C freezer for less than one year.

### 4.3. Nanoparticle Tracking Analysis

The size distribution of pancreatic cancer-derived EVs was examined by nanoparticle tracking analysis (NTA). Briefly, background measurements were performed with filtered PBS, which revealed the absence of any kinds of particles. EVs were diluted 1:1000 with PBS. After sample loading, five repeated measurements were recorded and then analyzed using a Nanosight NS300 with the NTA 3.0 software (Malvern Instruments, Malvern, UK).

### 4.4. Transmission Electron Microscopy

The morphology of pancreatic cancer-derived EVs was assessed by transmission electron microscopy (TEM). In brief, EVs were put onto formvar-carbon-coated electron microscopy grids for 10 min in a wet chamber. After brief blotting the grid edge with filter paper, the grids were placed on drops of 2% aqueous uranyl acetate for 1 min, removed, blotted again, and placed on H_2_O drops for 1 min, removed, and blotted at the edge. After 24 h of air drying, the grids were inspected using a transmission electron microscope (Zeiss EM 912 Omega at 100 kV, Darmstadt, Germany).

### 4.5. Western Blot

Cells and EVs were lysed using complete lysis M buffer supplemented with phosphatase inhibitor cocktail (Sigma, St. Louis, MO, USA). Protein samples were centrifuged at 14,000× *g* for 15 min at 4 °C. After centrifugation, the supernatants were transferred into new tubes and stored in a −80 °C freezer. Protein concentrations were measured by the BCA Protein Assay (Thermo Scientific™, Waltham, MA, USA). Protein Samples were prepared in Pierce™ LDS Sample Buffer (Thermo Fisher Scientific), boiled for 10 min at 70 °C. Ten to fifteen micrograms of lysates were loaded and run in SDS polyacrylamide gels. Gels were then transferred onto PVDF membranes. The membranes were blocked in blocking buffer at RT for 1 h and incubated with specific primary antibodies at 4 °C overnight (Appendix A). Blots were washed 3 × 5 min in PBST. Incubation with the appropriate horseradish peroxidase (HRP)-conjugated secondary antibodies was performed at RT for 1 h. Blots were again washed in TBST 3 × 5 min. Proteins were detected via chemiluminescence with SuperSignal™ West Pico PLUS Chemiluminescent Substrate (Thermo Fisher Scientific) using the Intas ChemoStar ECL Imager (Intas Science Imaging, Göttingen, Germany).

### 4.6. Flow Cytometry of EVs

Pancreatic cancer-derived EVs were incubated with polybead carboxylate microspheres (Polyscience, Niles, Illinois, USA) in PBS overnight at 4 °C. Samples were blocked with 2% BSA in PBS and centrifuged at 350× *g* for 2 min. The supernatants were discarded. The pellets were washed with PBS twice and resuspended in PBS. EVs were incubated with the anti-CD63 antibody at 4 °C for 20 min in the dark. The samples were analyzed by CytoFlex (Beckman Coulter, Indianapolis, IN, USA).

### 4.7. Flow Cytometry of Cells

For surface staining, cells were incubated with Human TruStain FcX™ (Biolegend, San Diego, CA, USA) on ice for 10 min. Then, conjugated fluorescent antibodies were added and incubated on ice for 20 min in the dark. After incubation, cells were washed in PBS twice and then analyzed by flow cytometry. For intracellular staining, surface antigen staining was performed as above. Afterwards, cells were washed and fixed in fixation buffer (Biolegend) for 20 min at RT. Cells were washed and permeabilized with permeabilization wash buffer (Biolegend). Cells were incubated with appropriate conjugated fluorescent antibodies in the dark for 20 min at RT. After incubation, cells were washed with permeabilization wash buffer twice and analyzed by flow cytometry. For each sample, 30,000–50,000 events were acquired using CytoFlex (Beckman Coulter). Data were analyzed using FlowJo software (Treestar Inc., Ashland, USA).

### 4.8. Mass Spectrometry of EVs

Fifty micrograms of pancreatic cancer-derived EVs were used for mass spectrometry analyses. First, EVs were denatured using 8 mol/L urea with protease inhibitor cocktail. Samples were centrifuged for 15 min at 17,000× *g* to remove debris. The protein concentration was determined using the Direct Detect^®^ Spectrometer (Darmstadt, Germany). Then, 50 μg per sample were transferred into a new 1.5-mL tube. Samples were reduced using 5 mM DTT at 25 °C for 1 h and incubated with 40 mM chloroacetamide in the dark for 30 min. This was followed by proteolytic digestion with lysyl endopeptidase (Lys-C) at an enzyme:substrate ratio of 1:75 at 4 °C for 4 h. The samples were diluted with 50 mM Triethylammoniumbicarbonate (TEAB) to achieve a final concentration of urea ≤2 M. A subsequent digestion with trypsin at an enzyme:substrate ratio of 1:75 and incubation at 25 °C overnight was performed. The digestion was terminated with formic acid at a final concentration of 1%. After digestion, the peptide mixtures were desalted using an in-house made StageTip per sample (containing 2 layers of SDB-RPS discs). StageTips were equilibrated as follows: 20 μL of methanol, centrifugation at 2600 rpm for 1 min; 20 μL of 0.1% formic acid in 80% acetonitrile, centrifugation at 2600 rpm for 1 min; 20 μL of 0.1% formic acid in water, centrifugation at 2600 rpm for 1.5 min; 20 μL of 0.1% formic acid in water, centrifugation at 2600 rpm for 2 min. The samples (acidified with formic acid) were centrifuged at full speed for 5 min and then loaded onto the equilibrated StageTips. After centrifugation at 2600 rpm for 5 min, the StageTips were washed according to the following protocol: 30 μL of 0.1% formic acid in water, centrifugation at 2600 rpm for 3 min; 30 μL of 0.1% formic acid in 80% acetonitrile, centrifugation at 2600 rpm for 3 min. The last wash step was performed twice. Finally, the StageTips were dried completely with a syringe and kept at 4 °C until LC-MS analysis. Nano LC-MS was performed using a gradient for 150 min and analyzed using the MAXQuant (Max Planck Institute of Biochemistry, Martinsried, Germany) and Perseus software (Max Planck Institute of Biochemistry, Martinsried, Germany).

### 4.9. In Vitro EVs’ Uptake Assay

Pancreatic cancer cell-derived EVs were isolated as described above. The PKH67 Fluorescent Cell Linker kit (Sigma, St. Louis, MO, USA) was used to label EVs according to the manufacturer’s instruction. In brief, after the first round of ultracentrifugation, the supernatant was discarded, and the pellet of EVs was resuspended in 750 μL of Diluent C. One microliter of PKH67 dye was dissolved in 250 μL of Diluent C. EVs and PKH67 dye were mixed gently and incubated at RT for 5 min. Nine milliliters of PBS with 1% BSA were added to bind excess PKH67 dye. EVs were ultracentrifuged at 100,000× *g* for 70 min at 4 °C and washed twice with PBS by ultracentrifugation. The PKH67-labeled EVs were then resuspended in PBS. NK cells were incubated with PKH67-labeled EVs for 24 h and put on polysine adhesion slides for 30 min at 37 °C. After fixation and permeabilization, NK cells were stained with DAPI. Uptake of PKH67-labeled EVs by NK cells was visualized by confocal microscopy.

### 4.10. In Vivo Distribution of EVs

To study the *in vivo* distribution of pancreatic cancer-derived EVs, EVs were fluorescently labeled as described above. Animal experiments were conducted according to protocols approved by the responsible national and local authority (81-02.04.2018.A139, LANUV NRW, approved on 20 September 2018). PKH67-labeled EVs were administered into the tail vein of two healthy 4–6-week-old NSG mice. One NSG mouse was injected with PBS as a negative control. Twenty four hours after injection, mice were euthanized. Organs were dissected and embedded in Tissue-Tek O.C.T.TM. Then samples were frozen and stored at −80 °C. For immunofluorescence, 10 μm of O.C.T.TM tissue cryosections were stained with DAPI. The distribution of PKH67-labeled EVs was analyzed by confocal microscopy.

### 4.11. In Vitro NK Cell Cytotoxicity Assay

L3.6pl cells (2 × 10^5^) were plated with NK cells (effector:target = 5:1) in a 6-well plate. All wells contained 100 IU/mL rhIL-2 with 50% of NK cell medium and 50% of L3.6pl culture medium. After co-culture for 24 h, plates were washed with DPBS three times. Adherent cells were harvested and counted for subsequent experiments.

### 4.12. Sphere Formation Assay

L3.6pl cells were seeded as a single-cell suspension at a concentration of 2000 cells/well in 6-well ultra-low attachment plates (Corning Inc., Corning, NY, USA) in DMEM/F12 medium (Gibco) supplemented with 20 ng/mL EGF (Peprotech, Rocky Hill, NJ), 20 ng/mL bFGF (Peprotech, Rocky Hill, NJ), 5 µg/mL insulin (Sigma-Aldrich, MO, USA) and 1× B27 (Gibco). After 7 days, spheres were counted under microscopy. Spheres were collected for mRNA extraction and flow cytometric analysis.

### 4.13. RNA Isolation, cDNA Synthesis, and qRT-PCR

Total RNA was isolated from cultured cells according to the manufacturer’s instructions using the RNeasy Mini Kit (Qiagen, Hilden, Germany). RNA was reverse transcribed according to the manufacturer’s protocol using the High-Capacity cDNA Reverse Transcription Kit (Applied Biosystems, ThermoFisher Scientific, Waltham, MA, USA). The synthesized cDNA was then assessed for gene expression using the Fast SYBR green master mix (Invitrogen) with QuantStudio 7 flex (Applied Biosystems, Thermo Fisher Scientific, USA).

### 4.14. Human Studies

All human tissue and blood samples were collected ethically, and their research use was according to the terms of the informed consent (BIOMASOTA, ID: 13-091, approval in May 2016). Human peripheral blood samples were acquired from healthy subjects and PDAC patients at University Hospital of Cologne between October 2016 and December 2018. All patients with PDAC were pathologically confirmed. Blood was obtained and centrifuged at 2000 rpm for 10 min and 4000 rpm for 10 min at RT. Serum samples were aliquoted and stored in the −80 °C freezer. Serum EVs were isolated as described above in the “EVs’ Isolation” section. The EVs’ TGF-β1 levels in serum were determined using the TGF beta-1 Human/Mouse Uncoated ELISA Kit (Invitrogen, ThermoFisher Scientific, Waltham, MA, USA) and analyzed with an ELISA microplate reader at 450 nm.

### 4.15. Statistical Analysis

Differences between two groups were tested by the two-tailed Student’s *t*-test. All statistical analyses were made using GraphPad Prism 8 (GraphPad Software, Inc., San Diego, USA). Data were considered statistically significant when the *p*-value was smaller than 0.05. 

## 5. Conclusions

In this study, we showed that tumor derived-EVs are responsible for pre-metastatic niche formation in the liver for PDAC. The inhibitory effects of pancreatic cancer-derived EVs on NK cells represent a mechanism allowing metastatic tumor cells to escape from NK cell immune surveillance in the pre-metastatic niche. We also demonstrate that TGF-β1 in serum EVs was significantly increased in patients with PDAC. In conclusion, these findings emphasize the immunosuppressive role of pancreatic cancer-derived EVs and provide new insights into our understanding of NK cell dysfunction regarding pre-metastatic niche formation for PDAC.

## Figures and Tables

**Figure 1 cancers-11-00874-f001:**
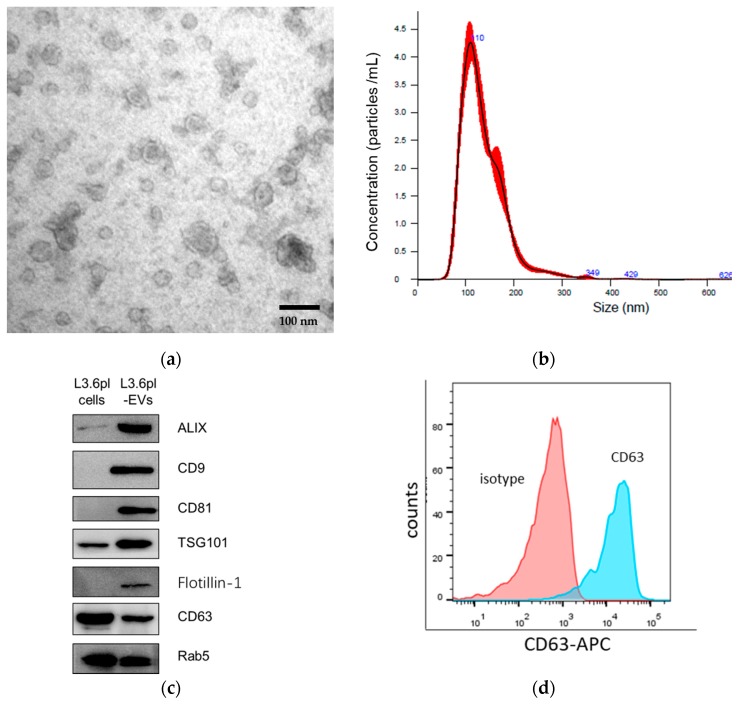
Characterization of pancreatic cancer cell-derived extracellular vesicles (EVs). (**a**) The representative image of pancreatic cancer-derived EVs by transmission electron microscopy (TEM). Scale bar, 100 nm. (**b**) The size of pancreatic cancer-derived EVs was determined by nanoparticle analysis (NTA). The size range was 136.1 ± 47.3 nm. (**c**) The expression of exosomal markers ALIX, flotillin-1, TSG101, CD9, CD63, CD81, and Rab5 for L3.6pl-derived EVs and parental cells was determined by Western blotting. (**d**) The expression of CD63 on L3.6pl-derived EVs coupled to carboxylate beads was analyzed by flow cytometry.

**Figure 2 cancers-11-00874-f002:**
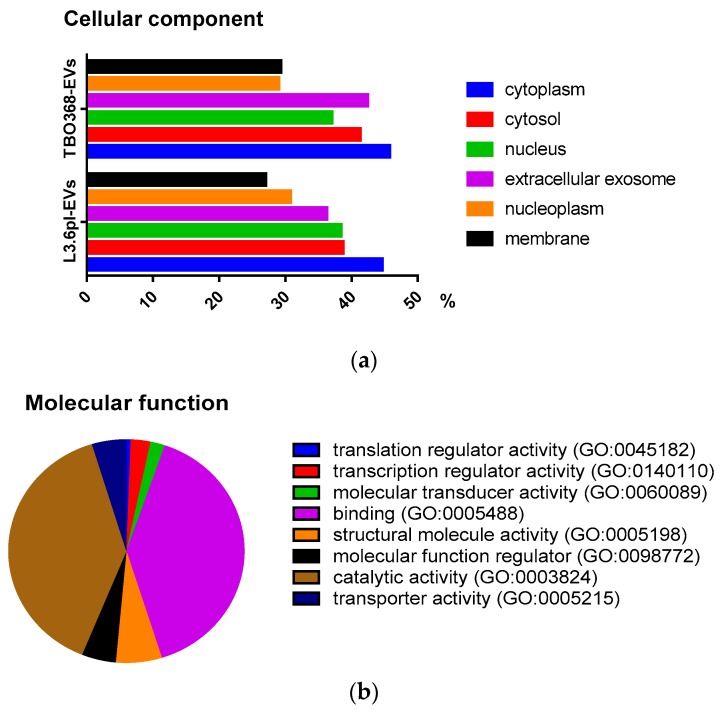
Proteomic analysis of pancreatic cancer-derived EVs. (**a**) The cellular component of proteins in TBO368-derived EVs and L3.6pl-derived EVs. (**b**) The molecular function of identified proteins in pancreatic cancer-derived EVs. (**c**) The biological process of identified proteins in pancreatic cancer-derived EVs.

**Figure 3 cancers-11-00874-f003:**
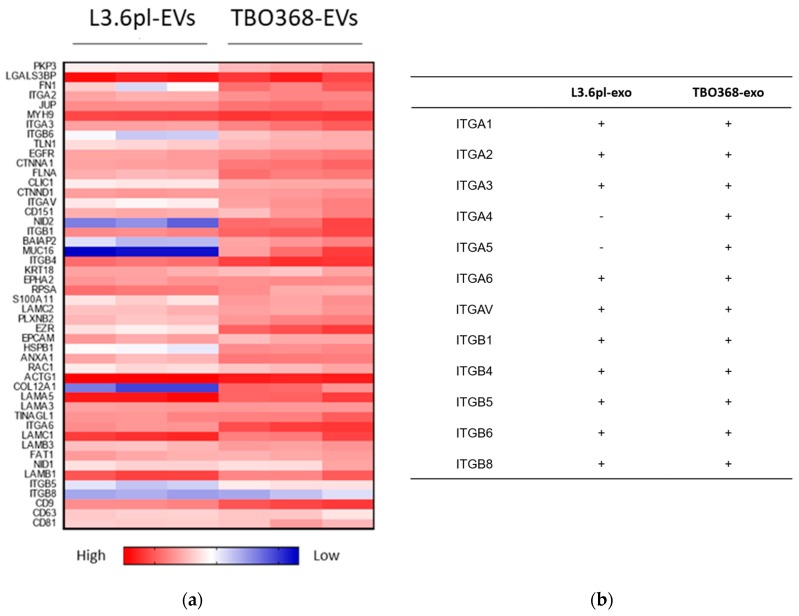
Pancreatic cancer-derived EVs carry adhesion molecules. (**a**) Heatmap of adhesion molecules in L3.6pl-derived EVs and TBO368-derived EVs, exosomal markers CD9, CD63, and CD81 as internal references. (**b**) Integrins in L3.6pl-derived EVs and TBO368-derived EVs. (**c**) Western blot analysis of ITGAV in L3.6pl-derived EVs. (**d**) Analysis of liver injected with PKH67-labeled L3.6pl-derived EVs (green) by confocal microscopy. Nuclei were stained with DAPI (blue).

**Figure 4 cancers-11-00874-f004:**
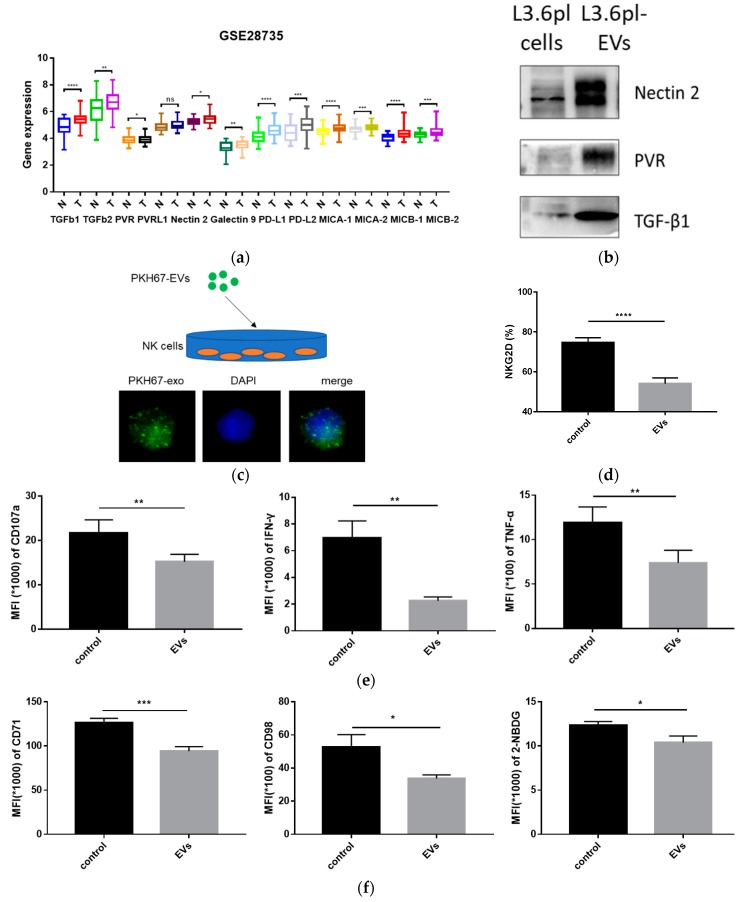
Pancreatic cancer-derived EVs impair natural kill (NK) cell function. (**a**) Relative mRNA expression of representative immune regulatory factors in tumor tissues (T) and non-tumor tissues (N) in pancreatic cancer from the GSE28735 dataset, *n* = 45. (**b**) The expression of nectin-2, PVR, and TGF-β1 was determined by Western blotting in L3.6pl-derived EVs and L3.6pl cells. (**c**) Analysis of pancreatic cancer-derived EVs’ uptake by NK cells using confocal microscopy. L3.6pl-derived EVs were stained with PKH67 (green) and incubated with NK cells for 24 h. The nucleus was labeled with DAPI (blue). (**d**) NK cells were treated with PBS or L3.6pl-derived EVs for 24 h. The percentage of NKG2D-positive NK cells was analyzed by flow cytometry. (**e**) NK cells pre-treated with PBS or L3.6pl-derived EVs were co-cultured with L3.6pl cells at a 1:1 ratio for 5 h. The mean fluorescence intensity (MFI) of CD107a (left), IFN-γ (middle), and TNF-α (right) in NK cells was analyzed by flow cytometry. (**f**) NK cells were treated with PBS or L3.6pl-derived EVs for 24 h. NK cells were then analyzed by flow cytometry to determine the MFI of CD71 (left) and CD98 (middle) and 2-NBDG incorporation (right). Data are the means ± SD of four experiments. ns, no significant difference, * *p* < 0.05, ** *p* < 0.01, *** *p* < 0.001, **** *p* < 0.0001 by Student’s t test.

**Figure 5 cancers-11-00874-f005:**
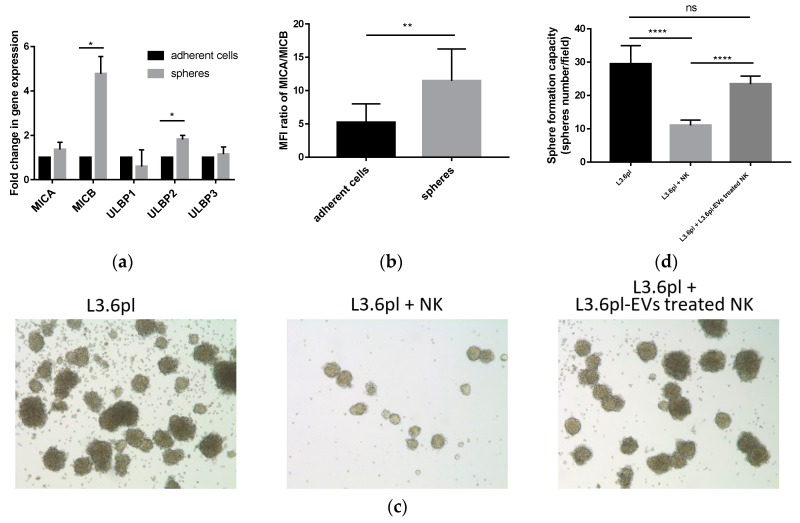
Pancreatic cancer-derived EVs suppress NK cell cytotoxicity against cancer stem cells (CSCs). (**a**) Gene expression of NKG2D ligands in adherent cells and spheres using qRT-PCR. Data were normalized to GAPDH and presented as fold change in comparison with genes in adherent cells. (**b**) The MFI of MICA/MICB in adherent cells and spheres was determined by flow cytometry. (**c**) Representative images of tumor spheres without NK cell killing (left), tumor spheres after untreated NK cell killing (middle), and tumor spheres after L3.6pl-derived EVs pre-treated NK cell killing (right). (**d**) The number of tumor spheres without NK cell killing, tumor spheres after untreated NK cell killing, and tumor spheres after L3.6pl-derived EV-pre-treated NK cell killing. Data are the means ± SD of four experiments. ns, no significant difference, * *p* < 0.05, ** *p* < 0.01, *** *p* < 0.001, **** *p* < 0.0001 by Student’s t test.

**Figure 6 cancers-11-00874-f006:**
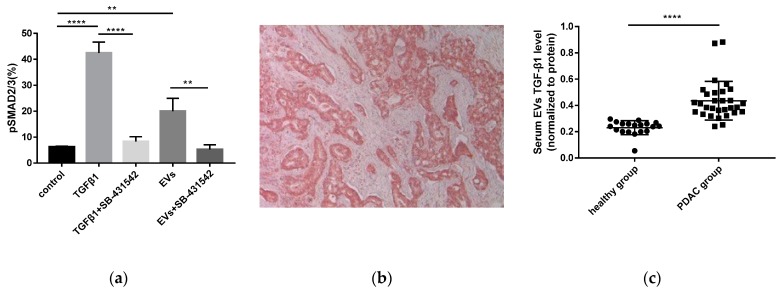
Pancreatic cancer-derived EVs inhibit NK cell function through the TGFβ-Smad2/3 signaling pathway, and serum EVs’ TGF-β1 is increased in pancreatic cancer patients. (**a**) After co-culture with TGF-β1 or L3.6pl-derived EVs in the presence or absence of SB-431542, the phosphorylation level of SMAD2/3 in NK cells was measured by flow cytometry. Data are the means ± SD of four experiments. (**b**) The immunohistochemistry (IHC) result showed TGF-β1 overexpression in tumor tissue of PDAC. Magnification: 100×. (**c**) The amount of TGF-β1 per gram of serum EVs in the healthy control (*n* = 19) and PDAC group (*n* = 30). ns, no significant difference, * *p* < 0.05, ** *p* < 0.01, *** *p* < 0.001, **** *p* < 0.0001 by Student’s t test.

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
