# Peer review of "Tumor-Derived Extracellular Vesicles Inhibit Natural Killer Cell Function in Pancreatic Cancer"

_cancers, 2019, doi:10.3390/cancers11060874_

Round 1

Reviewer 1 Report

The paper deals with a process of general relevance in oncology, the preliminary mechanisms of metastasis induced by aggressive pancreatic cancers. The paper starts from solid evidence established during the last few years, introducing a key role played by the extracellular vesicles released from cancer cells. This point is new, convincing and original. In my opinion, therefore, the paper deserves to be published.

The only problems I have found concern the nature of the critical EVs. Their analysis have fund an average size of 140 nm,  too much for exosomes which are reported to span from 50 to 100 nm. In addition, the images shown document a large heterogeneity of the vesicles, some of which are really small.

Taking into account that the specificity of exosomes with respect to the other type of EVs, the ectosomes is not black and white but quantitative, I strongly recommend to consider the analyzed vesicles as mixtures of the two types, to be defined only with the EV definition. The rest of the paper is certainly convincing.

Author Response

Dear Reviewers and Editor,

We appreciated a lot for your advice and guidance. As you have mentioned about the size of EVs and exosomes. A recent study (Dennis K Jeppesen et al, Cell, 2019) reported the size of exsomes could be ranged from 40nm to 150 nm; in contrast, the microvesicles with 150-1000 nm large were termed as extracellular vesicles. Our NTA analysis showed that the median size of EVs is 110 nm, and range with 136.1±47.3 nm. To some extend, this size is a little bit larger for exsomes . and we knew that it could not obtain pure exosomes with our current isolation methods. Therefore, it is not surprisingly to observe approximately 2-3 subpopulations of microvesicles in the TEM images, displaying the heterogeneity of the vesicles. We also attached the original image in the supplementary data. Based on both TEM and NTA results, we eventually only used the term as ‘Extracellualr Vesicels (EVs)’ to describe these secreted nanoparticles.

Reviewer 2 Report

The work submitted by Zhao et al aims to use Cancer-derived EVs to inhibit NK cells in pancreatic cancer.

However some questions need to be addressed:

1) Introduction:

- Authors should also report that tumour derived EVs might be used to deliver therapeutic agents to the tumour sites

- The aim of the study should be shortly highlighted in the introduction

2) Materials and Methods:

-Authors should specify how many cells have been used prior to start the EV isolation

- Any info about the storage period of EV-preparations at -80 is provided

-The paragraph related to the in vitro uptake study is incomplete, more details have to be provided

- Western blotting analysis need more clarifications about the all used antibodies an their working dilutions

-Animal permit is missing

-Any info about the health status of mice is provided

-Any info about how many animals have been used 

3) Results:

- Western blotting analysis need a proper negative control to exclude the presence of intracytoplasmatic contaminate.

- Authors evaluated the biodistribution by labelling cancer derived EVs with PKH67. However it is not clear how the labelling has been performed and which concentration has been used. This must be specified. Furthermore how the authors checked the fluorescence of the produced EV-formulations prior their injection in mice?Why the in vivo biodistribution has been checked in NSG mice? On the top of that a proper control tissue for the immunofluorescence study is missing

Author Response

Dear Reviewer and Editor,

we appreciated for the advice and guidance to improve our work. Below is our response to the comments: 

The work submitted by Zhao et al aims to use Cancer-derived EVs to inhibit NK cells in pancreatic cancer. However some questions need to be addressed:

1) Introduction:

- Authors should also report that tumour derived EVs might be used to deliver therapeutic agents to the tumour sites

Response : Thank you for the advice. We now update the part of ‘tumour derived EVs application of delivering therapeutic agents to the tumour sites’  in the introduction of this manuscript .

- The aim of the study should be shortly highlighted in the introduction

Response : We appreciated for the comments and highlighted the aim of the study in the introduction in revision.

2) Materials and Methods:

-Authors should specify how many cells have been used prior to start the EV isolation

Response : For EVs preparation from cell culture supernatants, pancreatic cancer cells with a confluency of 70-80%, around 1.5 x 107cells, were washed with DPBS for 3 times and were cultured in 25 mL of medium supplemented with 10% EVs-free FBS for additional 24 hours in a T-175 flask. We have clarified this information in the method.

- Any info about the storage period of EV-preparations at -80 is provided

Response : We store EVs for less than one year. We have described this information in the method.

-The paragraph related to the in vitro uptake study is incomplete, more details have to be provided

Response : Thank you for the advice to improve our work. In brief, after the first round of ultracentrifugation, the supernatant was discarded and the pellet of EVs was resuspend in 750 μL of Diluent C. 1 μL of PKH67 dye was dissolved in 250 μL of Diluent C. EVs and PKH67 dye were mixed gently and incubated at RT for 5 min. 9 ml of PBS with 1% BSA was added to bind excess PKH67 dye. EVs were ultracentrifuged at 100,000 g for 70 min at 4 °C and washed twice with PBS by ultracentrifugation. The PKH67-labeled EVs were then resuspended in PBS.NK cells were incubated with PKH67-labeled EVs for 24 h and put on polysine adhesion slides for 30 min at 37 °C. After fixation and permeabilization, NK cells were stained with DAPI. Uptake of PKH67-labeled EVs by NK cells was visualized by confocal microscopy. We will include this in the method.

- Western blotting analysis need more clarifications about the all used antibodies an their working dilutions

Response :Thank you for your advice. We have add detail information of antibody applications in the supplementary data.

-Animal permit is missing

-Any info about the health status of mice is provided. -Any info about how many animals have been used.

Response : We have now updated the information of animal ethic permission and the mouse status and amount in the manuscript. In addition, we described in the revision as : 10 μL of PKH67-labeled EVs (around 10 μg) were administered into the tail vein of two healthy 4–6-week-old NSG mice. One NSG mouse was injected with PBS as a negative control.

3) Results:

- Western blotting analysis need a proper negative control to exclude the presence of intracytoplasmatic contaminate.

Response: We appreciated for the advice. We have considered that EVs are systhesized inside the cells. Therefore, they contain a lot of components from cytoplasma. However, for EVs isolation, we centrifuged our samples at low speed to delete cells and cell debris. Thus, there are almost no intracytoplasmatic contaminate which might influence our data.

- Authors evaluated the biodistribution by labelling cancer derived EVs with PKH67. However it is not clear how the labelling has been performed and which concentration has been used. This must be specified. Furthermore how the authors checked the fluorescence of the produced EV-formulations prior their injection in mice?Why the in vivo biodistribution has been checked in NSG mice? On the top of that a proper control tissue for the immunofluorescence study is missing

Response:For EVs labeling, we introduced our techniques in the method part. In brief, after the first round of ultracentrifugation, the supernatant was discarded and the pellet of EVs was resuspended in 750 μL of Diluent C. 1 μL of PKH67 dye was dissolved in 250 μL of Diluent C. EVs and PKH67 dye were mixed gently and incubated at RT for 5 min. 9 ml of PBS with 1% BSA was added to bind excess PKH67 dye. EVs were ultracentrifuged at 100,000 g for 70 min at 4 °C and washed twice with PBS by ultracentrifugation. The PKH67-labeled EVs were then resuspended in 100 μL  of PBS.We check PKH67-EVs under the immunofluorescent microscopy to make sure that EVs are successfully labeled with PKH67. Our EVs were isolated from human pancreatic cancer cells. Therefore, we used NSG mice, which are immunodeficient. We also check immunofluorescence of the liver from a NSG mouse injected with PBS. We included this detail in the supplementary data.

Round 2

Reviewer 2 Report

Authors improved the manuscript, however the introduction is still lacking about information related to the use of EVs as delivery systems for chemotherapeutic agents and biologics which is up to date at the moment (Jang SC, ACS Nano 2013, Saari et al JCR 2015, Andaloussi et al Nature Reviews 2013, Garofalo et al JCR 2019). Therefore I suggest to add these aspects in the introduction

Author Response

Dear Reviewers and Editor,

We appreciated a lot for your advice on adding new aspects in the introduction of the manuscript, which is indeed helpful to strength the content. Therefore, we updated a paragraph as below for your review:

As lipid bilayer membrane vesicles, EVs are ideal carriersfor drug delivery in cancer treatment [1,2]. Kamerkar S et al. has successfully modified exosomes to deliver short interfering RNA specific to KRAS mutation, which suppressed tumor growth in PDAC-bearing mice and significantly increased their overall survival [3]. Jang SC et al. synthesized nanovesicles by the breakdown of monocytes or macrophages, which had similar characteristics with the exosomes. These nanovesicles could deliver chemotherapeutics to inhibit tumor growth [4]. Saari H and the team have loaded Paclitaxel into cancer cell-derived EVs, which could bring the drug into prostate cancer cells and increase its cytotoxicity [5]. In addition, a study led by Garofalo et al. reported that human lung cancer cell-derived EVs-formulationscould specificallytarget the neoplasia. Administration of EVs with oncolytic virus alone or in combination with chemotherapeutics may serve as a novel strategy to treat cancer [6].

Above part is also integrated into the whole revised manuscript.  

1.         Luan, X.; Sansanaphongpricha, K.; Myers, I.; Chen, H.; Yuan, H.; Sun, D. Engineering exosomes as refined biological nanoplatforms for drug delivery. Acta Pharmacologica Sinica 201738, 754, doi:10.1038/aps.2017.12.

2.         S, E.L.A.; Mager, I.; Breakefield, X.O.; Wood, M.J. Extracellular vesicles: biology and emerging therapeutic opportunities. Nature reviews. Drug discovery 201312, 347-357, doi:10.1038/nrd3978.

3.         Kamerkar, S.; LeBleu, V.S.; Sugimoto, H.; Yang, S.; Ruivo, C.F.; Melo, S.A.; Lee, J.J.; Kalluri, R. Exosomes facilitate therapeutic targeting of oncogenic KRAS in pancreatic cancer. Nature 2017546, 498-503, doi:10.1038/nature22341.

4.         Jang, S.C.; Kim, O.Y.; Yoon, C.M.; Choi, D.S.; Roh, T.Y.; Park, J.; Nilsson, J.; Lotvall, J.; Kim, Y.K.; Gho, Y.S. Bioinspired exosome-mimetic nanovesicles for targeted delivery of chemotherapeutics to malignant tumors. ACS nano 20137, 7698-7710, doi:10.1021/nn402232g.

5.         Saari, H.; Lazaro-Ibanez, E.; Viitala, T.; Vuorimaa-Laukkanen, E.; Siljander, P.; Yliperttula, M. Microvesicle- and exosome-mediated drug delivery enhances the cytotoxicity of Paclitaxel in autologous prostate cancer cells. Journal of controlled release : official journal of the Controlled Release Society 2015220, 727-737, doi:10.1016/j.jconrel.2015.09.031.

6.         Garofalo, M.; Villa, A.; Rizzi, N.; Kuryk, L.; Rinner, B.; Cerullo, V.; Yliperttula, M.; Mazzaferro, V.; Ciana, P. Extracellular vesicles enhance the targeted delivery of immunogenic oncolytic adenovirus and paclitaxel in immunocompetent mice. Journal of controlled release : official journal of the Controlled Release Society 2019294, 165-175, doi:10.1016/j.jconrel.2018.12.022.
